# Transcranial magnetic stimulation induced early silent period and rebound activity re-examined

Mustafa Görkem Özyurt[1], Heidi Haavik[2], Rasmus Wiberg Nedergaard[2], Betilay Topkara[1], Beatrice Selen Şenocak[3], Mehmet Berke Göztepe[4], Imran Khan Niazi[2], Kemal Sitki Türker[1]*

1 School of Medicine, Koç University, Istanbul, Turkey, 2 Centre for Chiropractic Research, New Zealand College of Chiropractic, Auckland, New Zealand, 3 Frank H. Netter MD School of Medicine, Quinnipiac University, North Haven, CT, United States of America, 4 School of Medicine, Akdeniz University, Antalya, Turkey

* kturker@ku.edu.tr

**Citation:** Özyurt MG, Haavik H, Nedergaard RW, Topkara B, Şenocak BS, Göztepe MB, et al. (2019) Transcranial magnetic stimulation induced early silent period and rebound activity re-examined. PLoS ONE 14(12): e0225535. https://doi.org/10.1371/journal.pone.0225535

**Data Availability Statement:** All relevant data are available from the Open Science Framework (DOI: 10.17605/OSF.IO/N7TXJ).

## Abstract

Despite being widely studied, the underlying mechanisms of transcranial magnetic brain stimulation (TMS) induced motor evoked potential (MEP), early cortical silent period (CSP) and rebound activity are not fully understood. Our aim is to better characterize these phenomena by combining various analysis tools on firing motor units. Responses of 29 tibialis anterior (TA) and 8 abductor pollicis brevis (APB) motor units to TMS pulses were studied using discharge rate and probability-based tools to illustrate the profile of the synaptic potentials as they develop on motoneurons in 24 healthy volunteers. According to probability-based methods, TMS pulse produces a short-latency MEP which is immediately followed by CSP that terminates at rebound activity. Discharge rate analysis, however, revealed not three, but just two events with distinct time courses; a long-lasting excitatory period (71.2 ± 9.0 ms for TA and 42.1 ± 11.2 ms for APB) and a long-latency inhibitory period with duration of 57.9 ± 9.5 ms for TA and 67.3 ± 13.8 ms for APB. We propose that part of the CSP may relate to the falling phase of net excitatory postsynaptic potential induced by TMS. Rebound activity, on the other hand, may represent tendon organ inhibition induced by MEP activated soleus contraction and/or long-latency intracortical inhibition. Due to generation of field potentials when high intensity TMS is used, this study is limited to investigate the events evoked by low intensity TMS only and does not provide information about later parts of much longer CSPs induced by high intensity TMS. Adding discharge rate analysis contributes to obtain a more accurate picture about the characteristics of TMS-induced events. These results have implications for interpreting motor responses following TMS for diagnosis and overseeing recovery from various neurological conditions.

## Introduction

Transcranial magnetic stimulation (TMS) of motor cortex brings about a short-latency excitatory response in skeletal muscles known as motor-evoked potential (MEP) [1,2]. When

**Funding:** We would like to thank participants who attended this study. Also, we acknowledge the role of Chiropractic Center of New Zealand and Koç University School of Medicine for funding this study. We thank the Scientific and Technological Research Council of Turkey (TÜB?TAK) for funding Mustafa Görkem Özyurt under "2211-A National Scholarship Programme for PhD Students". We also thank Australian Spine Research Foundation (ASRF) for partially funding the project.

**Competing interests:** The authors have declared that no competing interests exist.

elicited during voluntary contraction, MEP is followed by a period of reduced electrical activity or silence, known as cortical silent period (CSP). The duration of this silent period is dependent on the stimulus intensity and can last as long as 300 ms [3–5]. Both intracortical and spinal mechanisms have been suggested to contribute to this silent period. The initial portion has been suggested to be due to a contribution from spinal mechanisms involving changes in motoneuron excitability, afferent input and recurrent inhibition [6,7]. The remainder of the silent period has been suggested to be due to inhibition of motor cortical output mediated by $GABA_B$ receptors (for review see Ziemann [8]). Based on these claims, the duration of the CSP has been claimed to reflect strength of inhibition within cortex [7,9,10] and MEP to indicate net excitability of corticospinal pathway [11]. TMS induced MEP and CSP, therefore, are proposed as indicators for evaluating functions of central nervous system in plasticity [12,13] and various neurological diseases in clinical applications including, amyotrophic lateral sclerosis (ALS), Parkinson's disease and stroke [14–17].

Precisely because the size of MEP and duration of CSP are being used in the literature as indicators reflecting excitability of cortical circuitries, it is imperative that we know for sure whether or not they reflect real phenomena. For MEPs and CSPs values to be useful, it is imperative that the methods used to record and analyze them must be valid. The classical methods, averaged surface EMG (SEMG) and peristimulus time histogram (PSTH), used to study these phenomena have been assumed to be valid [18]. Using these analysis methods, it is commonly assumed that peaks and troughs in an averaged response to a stimulus indicate synaptic excitation and inhibition, respectively (reviewed in Türker and Powers [19]).

However, it has long been recognized that peaks and troughs in response to an afferent stimulus can reflect not only direct synaptic effects, but also secondary effects arising from the discharge statistics of the pre- and postsynaptic cells [20]. Direct synaptic effects on spike probability lead to subsequent changes in probability from phase advanced or delayed spikes (*count-related errors*), followed by secondary and tertiary peaks and troughs due to synchronization of the spikes in relation to the stimulus (*i.e., the synchronization-related errors*) [19]. It has been shown in regularly discharging motoneurons in brain slices that the probability-based methods (averaged rectified SEMG and PSTH) contain significant errors for indicating underlying synaptic potentials [21]. The errors in estimation using probability-based analyses, however, have been shown to be minimized if discharge rate analysis, peristimulus frequency-gram (PSF), is also used on the same data [19]. This is since the PSF is made up of superimposition of individual discharge rate points and these points do not add onto one other. Therefore, PSF does not generate count and synchronization-related errors as shown to be embedded in the classical probability-based methods [19,21].

In our previous work, we used threshold TMS intensity to study MEP and CSP in a hand [22] and a leg muscle [23] and analyzed the data using both probability and discharge-rate based analyses methods. Using stimulus intensities around MEP threshold, both of our previous studies showed increased rate of single motor unit (SMU) discharge following MEP lasting for tens of milliseconds [22,23]. Similarly, TMS pulses induced long-lasting facilitation in alert non-human primates [24] and cat primary visual cortex [25] using discharge rate-based analysis methods.

The current study was, therefore, planned to further characterize the properties of not only the CSP but also the MEP and rebound activity using suprathreshold TMS at varying intensities. We will be questioning whether CSP may include a long lasting excitatory period due to increased rate of SMU discharge as reported [22,23]. Therefore, we aimed to closely examine the motor unit discharge characteristics to better understand the synaptic potentials evoked by TMS in tibialis anterior (TA) and abductor pollicis brevis (APB) motor units. We hypothesize that the CSP may not reflect an inhibitory postsynaptic potential (IPSP) only but include a

period of net excitatory postsynaptic potential (EPSP), induced by single-pulse magnetic stimulation of the motor cortex.

## Materials and methods

The Human Ethics Committee of Koç University approved the experimental procedure (2017.124.IRB2.038). Experiments were performed in the neurophysiology laboratory of Koç University on subjects who signed informed consent forms. Total of 24 subjects (14 male and 10 female) participated in this study. Subjects were in the 18–30 age groups, had no known neuromuscular disease, brain injury or nervous system related disorder, had normal body habitus and were not regularly using prescribed medication. The experiments were performed on TA muscle of right leg (in 20 subjects) or APB muscle of the right hand (in 4 subjects).

### Recording configurations and subject preparation

Software Spike2 7.20 (Cambridge Electronic Design, England) was used to perform data acquisition and offline analyses. CED 1902 Quad System MKIII amplifier and CED 3601 Power 1401 MKII DAC were used for recording. To record the activity of the muscle, SEMG and intramuscular EMG were used. Two of the standard SEMG electrodes (Ag/AgCl) were placed on either muscle after rubbing with sandpaper, cleaning with alcohol and applying electrode gel to reduce the impedance. The electrodes were placed 4 cm apart for optimal signal recognition for TA muscle [26] but they were 2 cm apart for APB muscle due to its smaller size. In addition, tip-active Teflon insulated silver fine-wire electrodes (75 μm in core diameter; Medwire, USA) were used as intramuscular EMG electrodes. Those sterile bipolar wire electrodes were placed in the muscles *via* 25 G surgical needles in between two SEMG electrodes. After insertion, the needle was removed immediately by leaving a pair of fish-hooked fine wires inside the muscle. Sterile lip clip was used as a ground electrode in all trials for both SEMG and intramuscular EMG [27]. Both SEMG and intramuscular EMG recorded at a sampling frequency of 20,000 Hz. The recordings were filtered with a cut-off frequency of 20–10,000 Hz for SEMG and 200–10,000 Hz for intramuscular EMG.

Isometric force during dorsiflexion of the right foot was measured using a linear strain gauge (Model LC1205-K020, A & D Co. Ltd., Tokyo, Japan: linear to 196 N). The foot was restrained, and subjects dorsiflexed the foot against a plate that was positioned parallel to the base of the foot. Force signals were amplified (x 1,000), filtered (DC-100 Hz), and sampled at 2,000 Hz using the same data acquisition system.

### Determination of the optimum stimulation configurations

Each subject sat comfortably on a chair. The experiment began with three brief (5 seconds) maximal voluntary isometric contractions. Maximal efforts were separated by approximately 60 seconds of rest to avoid fatigue. Then, subjects performed isometric contraction that resulted in the firing of one or two SMUs which were easily recognizable (**Fig 1A**). A transcranial magnetic stimulator (Magstim 200$^2$, Magstim Co., Whitland, UK) with double cone coil electrode (110mm Double Cone Coil) was used to stimulate the corresponding region of the left hemisphere for right TA or figure of eight coil (D70mm Alpha Coil) for right APB muscle. MEPs were measured as the peak-to-peak amplitude of non-rectified recordings for each trial to determine optimum stimulation region. To find that point, we located the motor cortex according to the center of the head which was determined using the midpoint of the nasion-inion line. For APB, the stimulation site on the head was found using suprathreshold stimuli around the C3 region according to the international 10–20 system [28]. The figure-of-eight-shaped coil was oriented 45-degrees relative to the nasion-inion line compared to midpoint. On the other hand, we placed the double cone coil around the midpoint of the nasion-inion

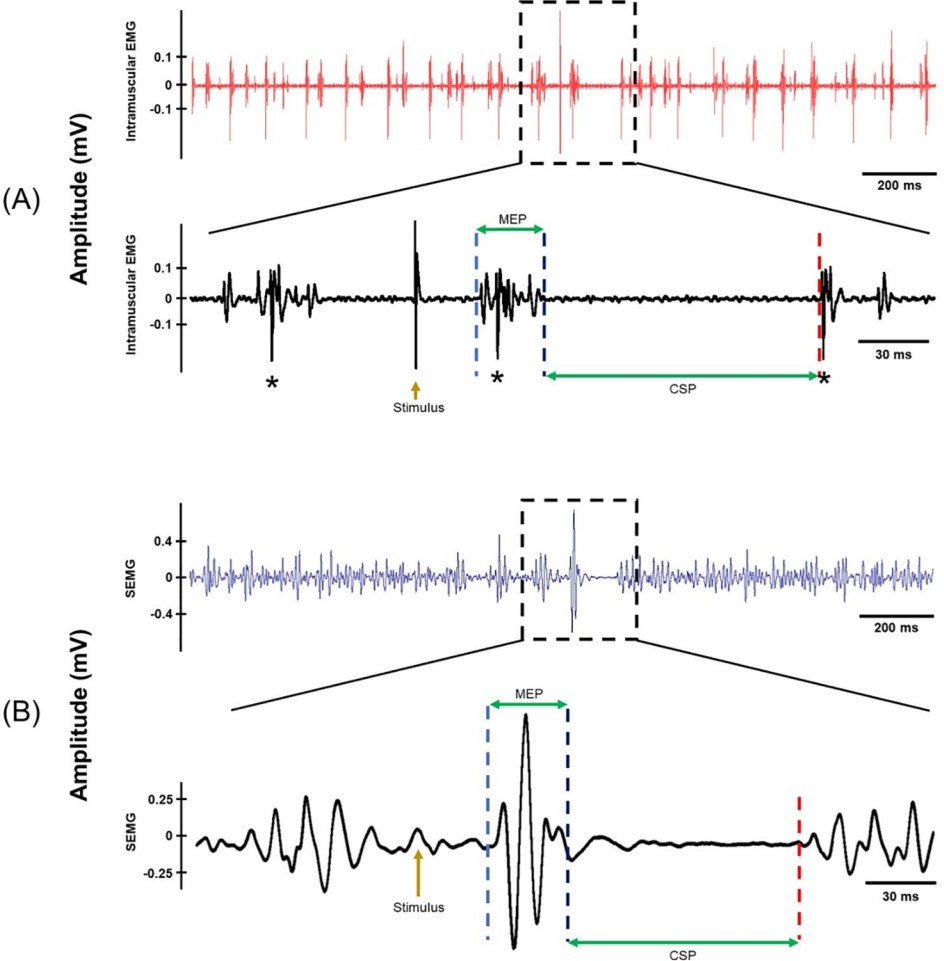

**Fig 1. A sample TA recording from a subject.** (A) Top trace is a typical intramuscular EMG recording where each line represents SMU potential. Figure below is the blown-up version of the intramuscular EMG where asterisk (*) shows SMU potentials. MEP and the CSP are indicated with horizontal arrows. (B) The figure above is the simultaneously recorded SEMG. Bottom figure is the closer view of the SEMG where MEP and the CSP are displayed. Higher intensity stimuli evoked complex field potentials at MEP period in intramuscular EMG recordings that made discrimination of SMU spikes very difficult.

line, but slightly angled in line with the C3 region, to stimulate the deeper side of the left motor cortex which is responsible for innervation of the right TA muscle.

The coil was positioned over the corresponding motor area of the left hemisphere as described. Single pulse TMS was delivered during weak isometric contraction for both muscles, separately. Initially, stimuli were delivered randomly between 4 and 6 seconds and at an intensity of approximately 150% of active motor threshold. The active motor threshold was defined as the minimum stimulus intensity at which 5 out of 10 consecutive stimuli evoke MEP of at least 100 $\mu$V in amplitude during weak muscle contraction in SEMG (**Fig 1B**). Then, the optimum location was marked and used throughout the experimental protocol.

## Experimental design

We performed several recording sessions per subject using suprathreshold TMS intensities. After deciding the optimal point, subjects performed weak muscle contraction throughout

stimulation period in all sessions. The desired voluntary contraction intensity was determined when only one or two SMUs were active and at least one of them could be identified by its amplitude (see **Fig 1A**). This information was provided to subjects as visual feedback to help them fire these low threshold SMUs regularly.

**Stimulation.** Stimulus intensity was determined according to the real-time triggering of SMU channel during on-line recording. TMS-triggered averaging window was monitored for the SMU channel where the number of occurrences of SMU potentials could be visualized during MEP period. For that purpose, simultaneous superimposition of the traces in the SMU channel was updated after each stimulus delivered and ***the number of SMUs that occur at the time period where MEP was expected to occur (marked MEP in Fig 1A)*** was determined. Various stimulus intensities that induced between 10 and 50 SMU potentials (at the time period where MEP was expected to occur) out of 100 TMS were used. This method was used to determine optimal stimulation intensity in order to track the full profile of the synaptic potentials. These intensities corresponded to an average strength of 33.56 ± 1.37% and 38.13 ± 1.17% (mean ± SEM) in terms of the maximum stimulator output for TA and APB, respectively.

**Force recording and sham stimulation.** Also, 4 subjects (1 female and 3 male) were recruited for twitch force recordings and sham TMS protocol for TA muscle using similar procedure. Sham TMS was applied using a homemade placebo coil that is similar in operation to the normal coil. The placebo coil provides minimal scalp sensation and an auditory click similar to that made by the normal coil, but without stimulating the cortex. The purpose for repeating the protocol with sham TMS was to determine whether other factors, such as the auditory click of the stimulator, contribute to the alteration of voluntary EMG observed after suprathreshold TMS. In addition to sham TMS, normal TMS were applied to these 4 subjects while twitch was recorded in order to determine the time course of synaptic potentials to compare with the muscle force generation time. We detected a visible twitch in 4 experiments (average twitch amplitude was; 0.93± 0.21% MVC).

**Stimulus and unit numbers.** Low intensity, suprathreshold stimuli of 541 ± 19 for TA and 320 ± 32 for APB were delivered to the motor cortex at each session. A total of 29 TA SMUs and 8 APB SMUs were recorded from 16 and 4 subjects respectively, using intramuscular EMG electrodes. For TA, only one SMU from each of 9 subjects were extracted. However, some other participants (7 out of 16) could clearly evoke different SMUs in each of the recording sessions in TA muscle. For the APB experiments, we analyzed two SMUs from each of the subjects (8 units from 4 different subjects in total).

## Data processing

To build both PSF and PSTH, electrical activity from the intramuscular fine wire electrodes was displayed and a shape of an individual motor unit action potential defined as a template in the Spike2 program. During the experiment and during off-line analysis, any spike whose shape matched this pre-established template, generated acceptance pulses in the program. The acceptance pulses from the discriminated units were used to construct PSTHs and PSFs around the time of stimulation using scripts of PSTH and PSF in Spike2. Basically, acceptance pulses around stimuli were placed into time bins to obtain PSTH. PSF was obtained by superimposing the instantaneous discharge rate values around the stimulus. In short, while PSTH is a histogram indicating the timing of occurrence of spikes against the stimulus, PSF is made up of superimposition of the instantaneous discharge rates of a selected unit around the time of the stimulus [29].

PSTH and SEMG records make significant errors in estimating synaptic potentials as stimulus-induced synchronization of spikes generate secondary and tertiary peaks and trough

related to autocorrelation function of the synchronous spikes rather than genuine synaptic events. Each discharge rate that make up of PSF record however represents an independent data point which indicates the membrane excitability at that time. Therefore, PSF does not include synchronous events and hence avoids the drawbacks of PSTH and SEMG [19].

Peak-to-peak amplitude and latency of MEPs were determined. For sustained contraction, SEMG analysis involved extraction of a period (-200 ms to +200 ms) from around each stimulus and averaging the signals. Intramuscular EMG analysis involved identification of SMU potentials using Spike2 algorithms. Analysis of each SMU involved extraction of a defined period from around each stimulus (-600 ms to +600 ms) followed by construction of a PSTH and PSF. Both PSF and PSTH were built with 0.1 ms binwidths.

Cumulative sums (CUSUMs) were then constructed from PSTH and PSF data. SMU firing probability (PSTH) and SMU discharge rate (PSF) responses were compared with SEMG responses. In addition, the effect of discharge rate on the duration of the CSP was investigated using PSF. CUSUMs were also calculated from averaged SEMG traces to illustrate subtle reflex responses [30]. While CUSUM for PSTH illustrates subtle but consistent changes in bin counts, CUSUM for PSF pinpoints subtle but consistent changes in discharge rate. Basically, CUSUM and CUSUM related analyses were performed in the following steps. First, the prestimulus average bin value in microvolts (for SEMG), in number of events (for PSTH), and in discharge rates (for PSF) was determined. Then the prestimulus average bin value was subtracted from each of the bin values in the entire analysis period (usually -200 to +200 ms period). Then the residual values left in each bin were integrated to obtain CUSUM. Therefore, CUSUM calculations simply sum up the differences of each bin value from the prestimulus average bin value and hence clearly indicate any subtle but continuous changes in the post-stimulus period that are not normally visible. Maximum deflection in the prestimulus CUSUM is used to build an error box which is then used to determine the significant poststimulus deflections [31,32]. Any post stimulus deflection that is larger than the largest prestimulus deflection and appears before the reaction time to this stimulus is considered as a genuine / significant response to stimulus [31].

Latency of a significant event in the CUSUM records is defined as the time of the turning point. Duration of a significant event is the horizontal distance between its first significant turning and the next significant turning. Strength of an event correspond to the vertical distance between the two CUSUM turning points and is expressed as a percentage of the maximum possible event [32]. If significant deflections are up-going, they are classified as 'excitation' and if they are down-going, as 'inhibition'. This is a conservative but accurate method for pinpointing genuine poststimulus deflections that underlie stimulus-induced changes in the motoneuron activity [21].

## Statistical analyses

We, firstly, searched the optimum TMS intensity which evokes several motor unit firings at the MEP period. For this purpose, we randomly adjusted stimulus intensity that generated SMUs at MEP period between 0 (sham) and around 50 SMUs per 100 stimuli, which was suggested to represent the injection of 0 to 5 mV EPSPs into regularly discharging motoneurons in previous studies [33,34]. Then, we calculated discharge rates of SMUs that are above prestimulus firing rate in the CSP region in all 29 SMUs of TA using nonparametric Wilcoxon matched-pairs signed rank test, after testing the normality using Shapiro-Wilk test. The duration of the CSP obtained with various analysis tools were compared using nonparametric Friedman test where multiple comparisons were corrected using Dunn's test. We have also checked if the background discharge rates in PSF influence the duration of CSP using

regression analysis. Statistical significance was set at p<0.05. All statistical analyses were performed using GraphPad Prism 7.

## Results

### Optimum stimulus intensity to trace synaptic potentials

To decide which intensity that could correctly trace and identify the event clearly, four different TMS intensities were used that generated 50, 30, 20 and 10 SMU occurrences at the MEP period out of 100 TMS pulses as well as control stimulation using sham coil in TA (**Fig 2**). The highest intensity (50 SMU occurrences out of 100 stimuli at the MEP period) stimulus evoked a large MEP but was too strong to allow us to trace the post-MEP events as seen in **Fig 2A**. However, the second highest stimulus intensity (generating 30 SMU occurrences for every 100 TMS) clearly traced the post-MEP events (**Fig 2B**). The intensity that induced 20 SMU occurrences out of 100 TMS also showed clear MEP and higher discharge rates during the post-MEP event period but relatively lower in number compared to the 30/100 stimulus intensity (**Fig 2C**). Other intensity (10/100) neither clearly showed the MEP nor the post-MEP events, so this was not the ideal intensity to investigate the post-MEP event characteristics (**Fig 2D**). Hence, due to better representation of MEP and following events in 20/100 and 30/100 intensities were used for investigating the MEP and post-MEP events. On the other hand, sham TMS application did not induce any response (**Fig 2E**).

### The cortical silent period and rebound activity

PSF, PSTH, and SEMG with their CUSUMs were also used to investigate late post-MEP events (**Fig 3**). PSF clearly showed that the units were firing at higher discharge rates at the CSP region than at the prestimulus region especially when stimulus intensities that delivered 20–30 SMU occurrences out of 100 stimuli. However, SEMG and PSTH analyses indicated that the electrical activity was lower during CSP hence suggesting an inhibitory period due to their definition for inhibition, i.e., lower number of spike events during a period compared with the prestimulus spike counts (**Fig 3A–3C, events marked as "1"**).

Again, in the classical probability-based analysis there was a rebound excitatory period (secondary peak) immediately after the CSP (**Fig 3B and 3C, marked as "2"**). Unlike the SEMG and PSTH, PSF results showed that the secondary peak was actually indicating an inhibitory event (**Fig 3A, indicator "2"**) as the discharge rates of spikes during this period were lower than the average prestimulus discharge rate (for definition of inhibition see and Kudina [35]). We observed this secondary inhibitory event clearly in 19 out of 29 SMUs in TA and 6 out of 8 SMUs in APB with duration of 57.9 ± 9.5 ms for TA and 67.3 ± 13.8 ms for APB.

### Pre-stimulus vs early post-stimulus discharge rate of motor units

We compared the number of units fired 2xSD above the background discharge rate at prestimulus and early poststimulus (CSP, **see Fig 3 part "1"**) region in TA muscle. Prestimulus time was selected as the time between -250 to 0 ms, while poststimulus time is the duration of the CSP which was calculated using PSF-CUSUM. The number of high frequency firings (i.e. above 2xSD) obtained at both regions was then normalized timewise to the 100 ms of duration. Shapiro-Wilk test revealed non-normal distribution for both prestimulus region (W = 0.8969, p = 0.0083) and CSP (W = 0.9013, p = 0.0106). A significantly higher firings during CSP was found in TA muscle (p = 0.0042, Wilcoxon matched pairs signed rank test) compared to prestimulus period (**Fig 4**).

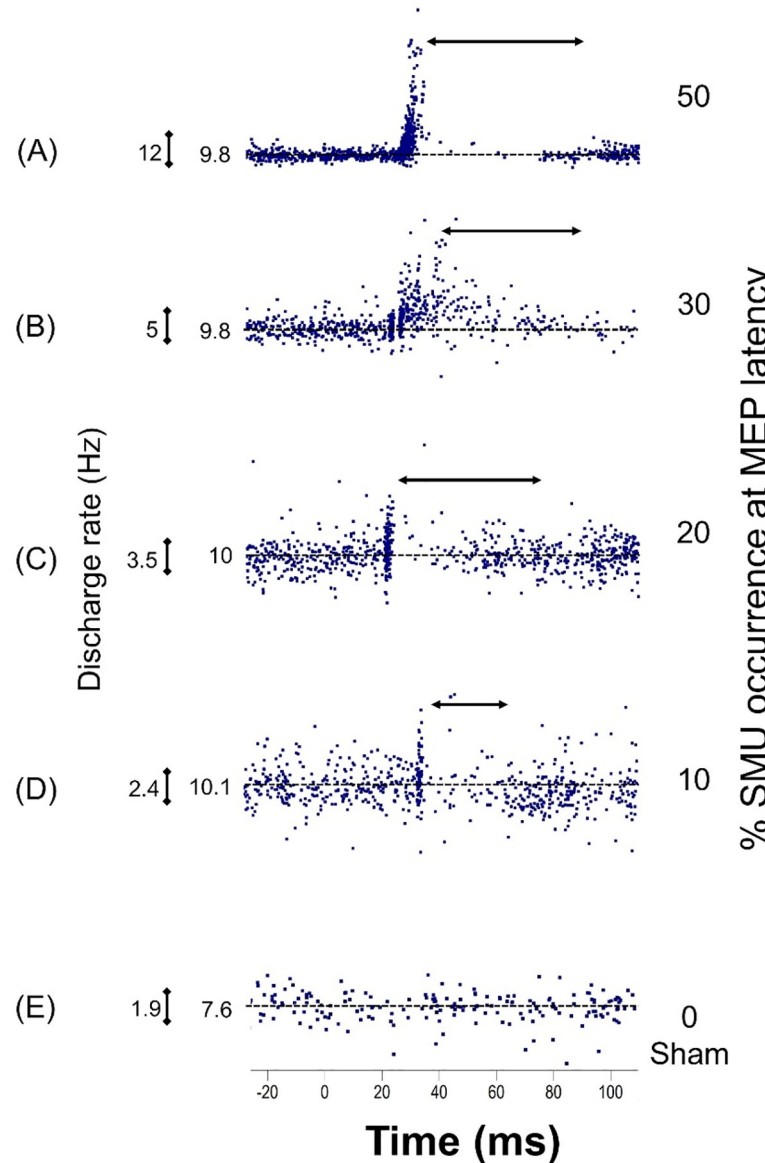

**Fig 2. Representation of discharge rates (PSF) in different TMS intensities in TA muscle.** The horizontal arrows indicate duration of the TMS induced CSP. Latency was calculated using PSTH-CUSUM and the duration using PSF-CUSUM. The intensities which evoked (A) 50, (B) 30, (C) 20, (D) 10-unit potentials at the MEP period per 100 stimuli, and (E) sham TMS without any MEP. Note that in 30/100 intensity, the motor unit responses during MEP clearly indicate I1 and I2 responses. In all cases, average the discharge rates of SMUs and scalebars that indicate calibrations in each trace are shown.

### Different analysis tools and duration of the early cortical silent period

We calculated the duration of the CSP using CUSUMs in all 29 units from 20 subjects for TA muscle (p = 0.0117, Friedman). The longest CSP durations was obtained in PSF (71.2 ± 9.0 ms) and PSTH (49.2 ± 3.9 ms) with insignificant difference (p>0.99, Dunn). On the other hand, the duration in SEMG (46.4 ± 7.2 ms) was significantly shorter compared to the duration that was calculated using PSF (p = 0.0175, Dunn) but not shorter than the duration in PSTH (p = 0.0543, Dunn).

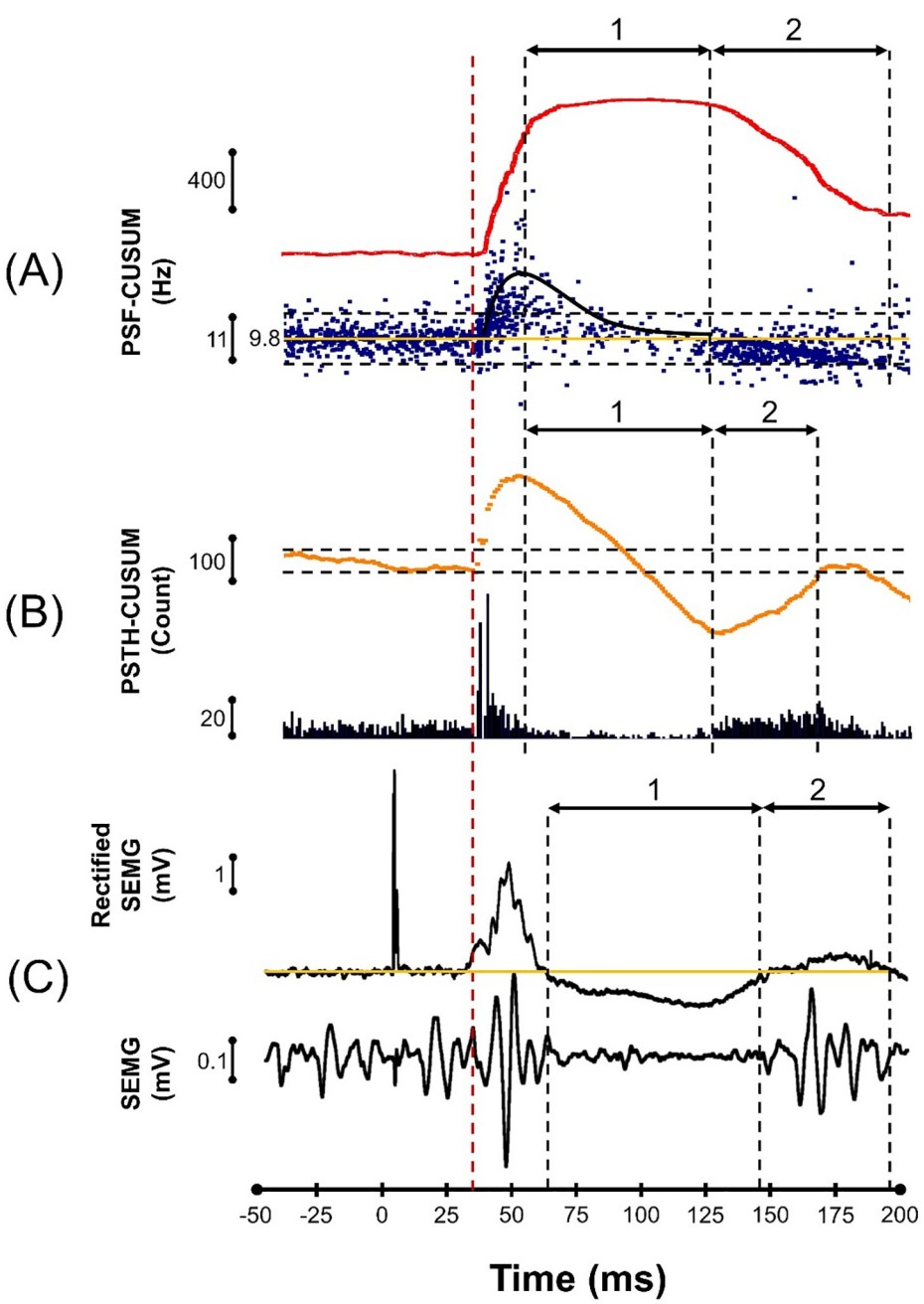

**Fig 3. Representation of MEP and early (1: CSP) and late (2: Rebound activity period) post-MEP events with a stimulus intensity that generated 30-motor unit occurrences at the MEP period per 100 stimuli in TA.** Black vertical dashed lines indicate the post-MEP event onsets and endpoints while red dashed line indicates MEP latency. Black horizontal arrows present CSP (1) and Rebound activity period (2). Early post-MEP period is the classical CSP and the late period is the classical 'Rebound Activity' that terminates the CSP. (A) PSF illustrates the frequency pattern of the unit together with its CUSUM (top trace). Horizontal dashed lines indicate 2 x SD according to the pre-stimulus firing rates and yellow line is average background discharge rate which was 9.8 Hz. (B) PSTH represents the firing probability of the unit, represented with its CUSUM (orange trace above). Dashed lines in CUSUM indicate the error box limits (see Methods). (C) Averaged-rectified SEMG-CUSUM response shows the MEP latency and reduced activity (black trace above).

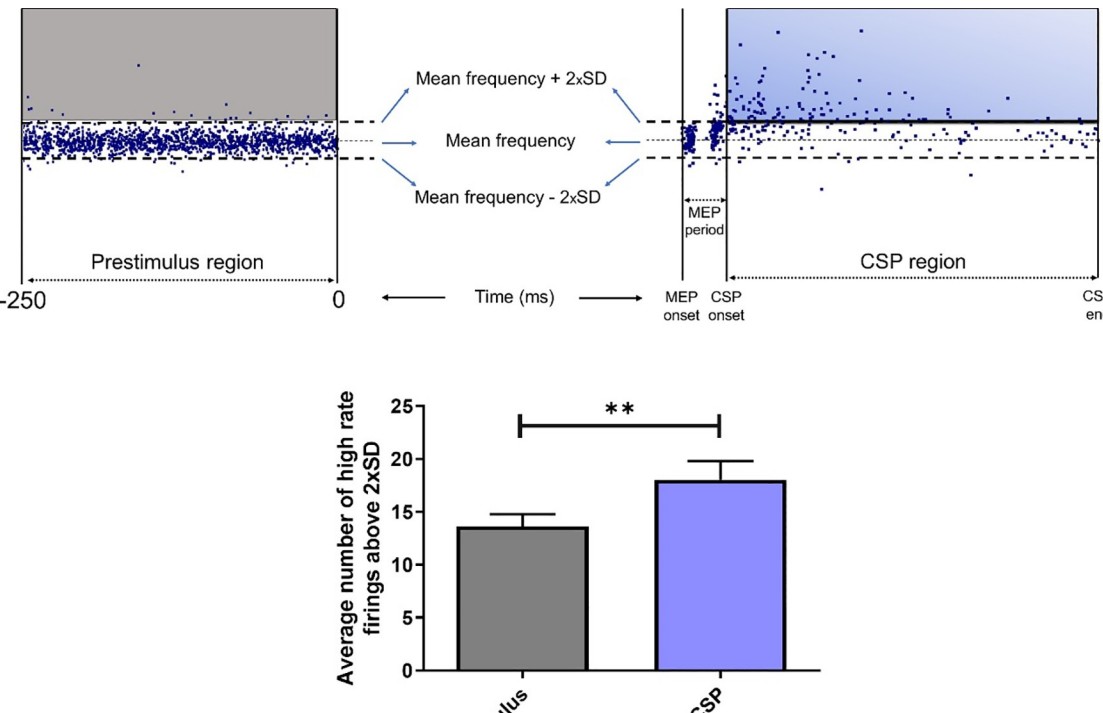

**Fig 4. Average number of units firing above 2xSD for different TMS intensities at prestimulus region and early poststimulus region (CSP) in TA muscle.** The number of high-frequency firings during prestimulus time of 250 ms (top left figure, units in gray rectangular shape) was compared with the average number of high-frequency firings at the CSP region (top right figure, units in blue rectangular shape) after normalization to 100 ms (see text for details). The average number of high-frequency events at prestimulus and CSP regions for 29 units are shown in gray and blue column at the bottom, respectively **p<0.01. Error bars are SEM and N = 29.

Similarly, in 8 of APB SMUs from 4 subjects, no significant difference was observed when duration of CSP calculated in PSF (42.1 ± 11.2 ms), PSTH (40.0 ± 8.0 ms) and SEMG (35.5 ± 8.2 ms) (p = 0.7943, Friedman and p>0.99 for all multiple comparisons, Dunn).

### Background firing rate and duration of the early cortical silent period

The relationship between the CSP and discharge rate of the units was investigated (Fig 5). Linear regression revealed a significant inverse relationship between the CSP and background discharge rate in PSF for both muscles (TA: p = 0.0005 and APB: p = 0.0221).

### Discussion

Several unique findings have been proposed in this study. Firstly, we found that background discharge rate of motor units significantly altered CSP duration. Secondly, even though the number of spike occurrences was lower than average pre-stimulus spike number during CSP, discharge rate of motor units was higher than the pre-stimulus discharge rate. Therefore, early CSP may not be an inhibitory period but may instead represent a net excitation induced by the TMS pulse and possibly contributed by both spinal and cortical mechanisms. Both TA and APB had similar CSP characteristics. Lastly, our findings suggest that the rebound activity at the end of the CSP indicates existence of a net IPSP, again possibly contributed by both spinal

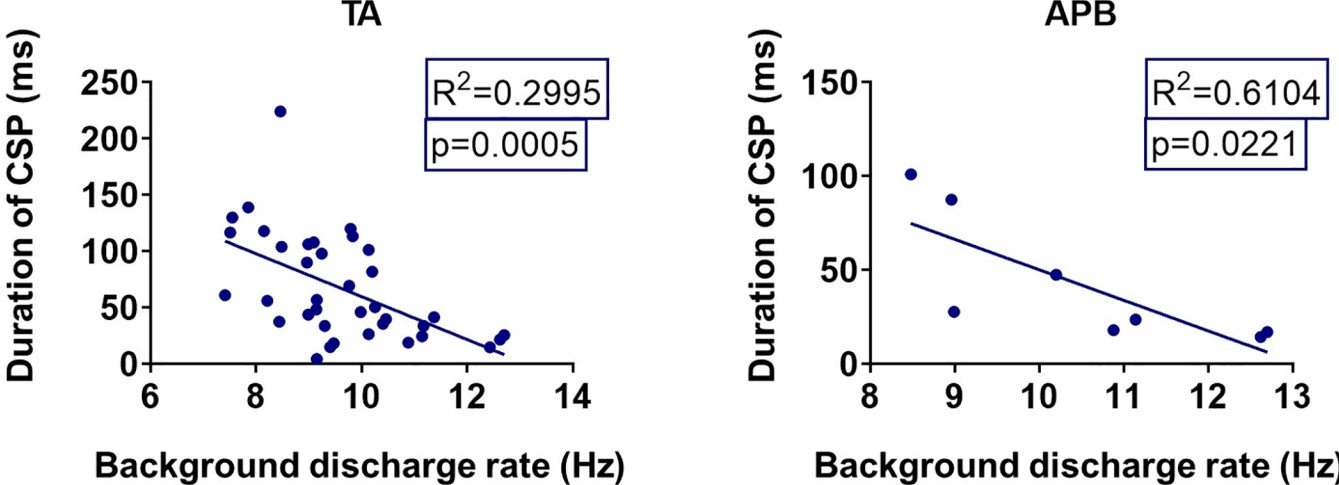

**Fig 5. The correlation between the background discharge rate of the SMUs and duration of the CSP measured using PSF-CUSUM.** Left figure shows the relationship for 29 units measured in TA muscle, while the figure on the right represents 8 APB units.

and cortical mechanisms. This study illustrates the importance of combining the techniques of SEMG, PSTH, and PSF to accurately characterize net synaptic events evoked by TMS.

## Background discharge rate of motor units significantly alters the duration of cortical silent period

It has been previously suggested that the duration of CSP is strongly influenced by stimulus intensity [36,37]. Therefore, standardization of some parameters is needed to determine reliable comparison between health and disease as MEP and CSP can vary with the exact stimulation point, level of stimulus intensity and prestimulus contraction level [37,38]. In the current study, while SEMG and PSTH analyses did not indicate any change in CSP duration with the level of prestimulus contraction, PSF displayed a significant decrease in CSP duration with an increase in the background discharge rate of motor units. This correlation may explain reported reduction in CSP duration in spastic patients since these patients display increased motor unit discharge rates especially on the less-affected side [39].

The duration of CSP we found in this study was shorter than CSP duration measured in classical studies for both upper and lower limb muscles [5,40–42]. The reason for this difference could be due to the lower stimulus intensity that was used in the current study. Both this study and a study by van Kuijk et al. [40] clearly showed that the duration of CSP is shorter when lower stimulus intensities are used. Therefore, to reduce the possibility of large *field potentials* at MEP and post-MEP event periods, which interferes with correct recognition of SMU spikes, we were limited to use lower stimulus intensities around 30–40% of maximum stimulator output. This has led to shorter CSP durations in the current study.

## Early cortical silent period may represent a net excitatory postsynaptic potential

In the literature, it has been suggested that both intra-cortical and spinal mechanisms contribute to the TMS induced CSP where spike occurrence is low. Reduced spike occurrences after a large MEP may not however indicate an inhibitory event as the MEP can cause phase advancement of spikes to an earlier time period hence generating a gap in discharge probability

immediately after MEP. Furthermore, when a motor unit fires at a given time it cannot fire for tens of milliseconds due to its afterhyperpolarization property [43,44].

While the classical analysis methods, SEMG and PSTH, identified the CSP with reduced spike activity; frequency analysis (PSF), on the other hand, suggested that CSP may represent a mixed event in which compound net excitation during CSP may be a combination of the falling phase of a large EPSP (rising phase of which induces the MEP) and some corticospinal inhibitory circuits that are activated by the TMS pulse. Similar long lasting excitatory periods have been described in animal studies where activity of single cortical neurons were recorded in response to TMS [24,25]. The rationale for suggesting CSP to represent a net excitatory event comes from the fact that the discharge rates of small number of spikes that fire during CSP were significantly higher than the average prestimulus discharge rate (**Fig 4**). This indicates that a low number of spikes following MEP may be due to being in the shadow of a large net EPSP that usually crosses the firing threshold during its rising phase. Threshold crossings during the falling phase of such a large net EPSP can only occur due to synaptic noise and only in rare occasions (**Fig 6**). This situation was observed in both TA and APB muscles. This finding suggests that the CSP may not be an inhibitory event only, instead, it may represent a combination of excitations and inhibitions resulting in a net EPSP.

Previously proposed mechanisms for the CSP, especially for its late part, have focused on cortical inhibition [45–47]. Inghilleri et al. [45] illustrated that the longest duration of CSP they observed was around 300 ms, and the first 50 ms was mediated by spinal circuitries upon brainstem stimulation. Similarly, paired TMS with epidural recordings of indirect waves (I-waves) showed no changes in I-waves in the first 50 ms of the CSP, whereas, these waves were reduced in the later parts of the CSP (after 100 ms) [46]. Moreover, there was no observable effect of Vigabatrin, a selective GABAergic drug, neither on the peripheral motor excitability

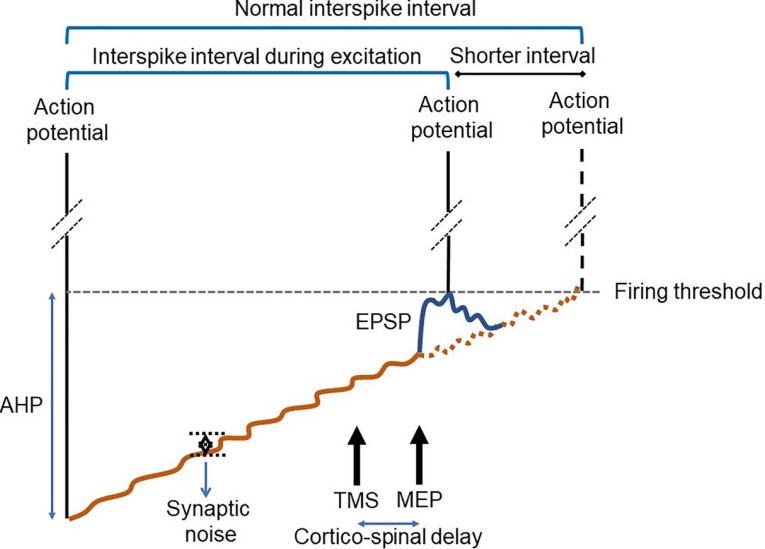

**Fig 6. Hypothetical motoneuron discharge to illustrate the effect of TMS induced net EPSP on ongoing action potentials.** Rising phase of the EPSP crosses firing threshold at most cases as it is larger than the synaptic noise and also is rapidly-rising. Threshold crossing by rapidly-rising phase of EPSP effectively brings action potentials that were to occur later to an earlier time (phase advance of spikes). This creates a period of low firing probability (cortical silent period; CSP) immediately after the rising phase of EPSP as spikes that were to fire in that period moved to occur earlier, generating MEP in the SEMG. Threshold crossing during falling phase of an EPSP is only possible when fast-rising phase fails to cross the threshold and the falling phase of the EPSP crosses the threshold with the help of an up-going synaptic noise. This is an extremely rare event and hence most of the threshold crossings will be achieved during the rising phase of an EPSP especially when EPSP is large.

nor on the early part of CSP [47]. This indicates that the late cortical component of the CSP is likely an inhibitory phenomenon mediated by GABA.

In detail, it has been argued that the cortical inhibition is presynaptic to the cortico-spinal neurons, rather than due to a decreased excitability of these cortico-spinal neurons [7,48,49]. Other neuropharmacological modulation in healthy subjects have suggested that the CSP reflects GABA$_B$-mediated intracortical inhibition [50–53]. Siebner et al. [50] observed a marked prolongation of the CSP during continuous intrathecal administration of high doses of the GABA$_B$ receptor agonist baclofen in a patient with generalized dystonia. Werhahn et al. [51] showed in healthy subjects that the ingestion of a single dose of tiagabine (which inhibits the uptake of GABA from the synaptic cleft) prolonged the TMS-induced silent period duration. Both groups argued that this prolongation must reflect increased intracortical inhibitory changes. This prolonged inhibition could reflect the changes in the later-onset event, suggested as the rebound activity (net IPSP) in this study.

Our suggestion about the net EPSP during CSP is made up of a contribution from the falling phase of MEP-induced EPSP and TMS-induced IPSPs in cortical and spinal networks. One of the contributing IPSPs may come from the Renshaw circuitries in the spinal cord [54] which may explain prolongation in CSP after GABA-specific drug administration that may cause enhanced/longer recurrent inhibition.

### Rebound activity may indicate an inhibitory postsynaptic potential

Rebound activity observed in this study marks the end of CSP in the classical studies [18,36,48,55]. As can be seen in **Fig 3**, reduction in the discharge rate during this period as indicated in PSF has been claimed to be a rebound excitation in PSTH and SEMG analyses. However, during this period of rebound activity, the discharge rate of motor units decreases below the level of prestimulus discharge rate (**Phase 2 in Figs 3 and 7**). PSF method clearly shows that the rebound activity following CSP is made up of synchronous occurrence of phase delayed spikes. The reason for the phase delay in spike timing is due to a net IPSP.

Several hypotheses can be put forward about the mechanism of this late inhibition (**Fig 7, see part "2"**). The effective inhibition may come from the activation of tendon organs, Renshaw circuitry, other spinal inhibitory circuits and/or cortical inhibition mediated by GABA$_B$. It is possible that muscle contraction that is initiated by TMS pulse can activate the tendon organs within the muscle. It is known that the tendon organ activation induces IPSP on homonymous muscles [56] and its latency (incorporating: electromechanical delay + afferent conduction time + synapses) is similar to the latency of the rebound activity described in this study even though the muscle twitch was rather weak due to smaller stimulus intensity. Our findings suggest that the net IPSP should be arriving at the muscle 75–100 ms after TMS pulse, therefore, GABA$_B$-mediated intracortical inhibition might also contribute to this late-onset inhibition as previously reported by epidural recordings [46] as well as other approaches show the late part of the IPSP is cortical origin [6,45,57].

The time delay for recurrent inhibition to show its effect should be several milliseconds following MEP and recurrent IPSP can last for about 40–50 ms on firing motoneurons [54]. At around the time onset of IPSP reported in this study, the effect of recurrent inhibition would wither away. Therefore, recurrent mechanism is unlikely to be the cause of the late-onset IPSP observed in the current study though it can be involved in the net EPSP during CSP, due to its timing.

In summary, this study proposed that the early part of the CSP may not be an inhibitory event; instead, it may represent a long-lasting net EPSP. Moreover, following CSP, motor units fired at lower discharge rates and therefore this period of rebound activity may represent

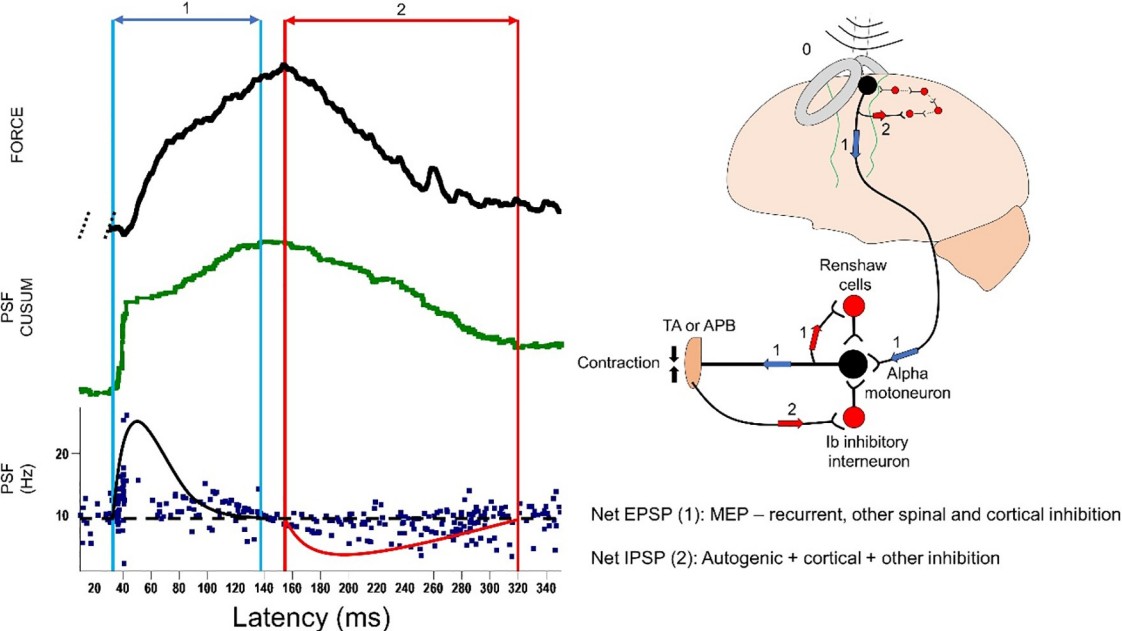

**Fig 7. The proposed mechanisms behind rebound activity (net IPSP).** About 30 ms after TMS pulse; a large and long-lasting net EPSP (MEP induced EPSP–recurrent inhibition and other inhibitions in the circuitry) is recorded from the muscle (CSP, 1). Following this net EPSP, a second PSP (net IPSP) develops on motoneurons (rebound activity), 2) which could be due to autogenic inhibition + intracortical inhibition + other unknown inhibitory and excitatory mechanisms. Left: Force trace and PSF as well as its CUSUM of a unit recorded from TA. Right: proposed mechanism responsible for the net EPSP and net IPSP. Black parabolic curve represents imaginary profile of TMS-induced net EPSP while red curve shows the imaginary profile of net IPSP. Blue and red arrows show durations of net EPSP and IPSP, respectively. Black horizontal dashed line in the PSF is the average background firing rate.

existence of a net IPSP. This study illustrated the complexity of the mechanisms underlying MEP, CSP and rebound activity which are confounded by the activation of several networks both in the cortical and spinal levels. What we record from the motoneurons as the *final end product* is a combined *net effect* of all these TMS-evoked spinal and supraspinal circuitries.

## Limitations of the study

Due to the problem with the field potential, we could not use high stimulus intensities. High stimulus intensities induced a response not only on the regularly discharging SMUs but also on other previously non-active motor units. Therefore, high-level TMS pulses generated a field potential (**Fig 1A**) at the MEP latency which made recognition of SMU potentials impossible. This limitation also restricted us from directly comparing the MEP and CSP sizes and durations with the literature as they have used much stronger stimulus intensities than this study. This means our analysis was constrained to the first 50 ms of the CSP, whereas CSP duration is typically in the order of several hundreds of milliseconds. Although GABA$_B$-mediated cortical inhibition is distributed within most of the corticospinal neurons, the inhibition evoked in the current study using low intensity TMS might have been relatively weak to express itself in the TMS-induced net response. However, we are satisfied to suggest that TMS stimulation within the limits of the current framework induces a long-lasting net EPSP which is followed by a net IPSP. We propose that stronger stimuli would increase the size of the EPSP and hence make it impossible to record action potentials in the falling phase of the net EPSP (see for instance a stronger intensity induced CSP in **Fig 2A**).

## Author Contributions

**Conceptualization:** Mustafa Görkem Özyurt, Heidi Haavik, Kemal Sitki Türker.

**Data curation:** Mustafa Görkem Özyurt, Rasmus Wiberg Nedergaard, Betilay Topkara, Beatrice Selen Şenocak, Mehmet Berke Göztepe, Imran Khan Niazi, Kemal Sitki Türker.

**Formal analysis:** Mustafa Görkem Özyurt, Betilay Topkara, Beatrice Selen Şenocak, Mehmet Berke Göztepe, Imran Khan Niazi, Kemal Sitki Türker.

**Funding acquisition:** Heidi Haavik, Kemal Sitki Türker.

**Investigation:** Mustafa Görkem Özyurt, Rasmus Wiberg Nedergaard, Kemal Sitki Türker.

**Methodology:** Mustafa Görkem Özyurt, Mehmet Berke Göztepe, Imran Khan Niazi, Kemal Sitki Türker.

**Project administration:** Heidi Haavik, Imran Khan Niazi, Kemal Sitki Türker.

**Resources:** Rasmus Wiberg Nedergaard, Kemal Sitki Türker.

**Supervision:** Kemal Sitki Türker.

**Validation:** Mustafa Görkem Özyurt, Betilay Topkara, Mehmet Berke Göztepe, Kemal Sitki Türker.

**Visualization:** Mustafa Görkem Özyurt, Kemal Sitki Türker.

**Writing – original draft:** Mustafa Görkem Özyurt, Beatrice Selen Şenocak, Kemal Sitki Türker.

**Writing – review & editing:** Mustafa Görkem Özyurt, Heidi Haavik, Rasmus Wiberg Nedergaard, Betilay Topkara, Beatrice Selen Şenocak, Mehmet Berke Göztepe, Imran Khan Niazi, Kemal Sitki Türker.

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
