## [Decision Letter · Decision Letter 0]

6 Aug 2019

PONE-D-19-18148

TRANSCRANIAL MAGNETIC STIMULATION INDUCED SILENT PERIOD AND REBOUND ACTIVITY RE-EXAMINED

PLOS ONE

Dear Prof. Turker,

Thank you for submitting your manuscript to PLOS ONE. After careful consideration, we feel that it has merit but does not fully meet PLOS ONE’s publication criteria as it currently stands. Therefore, we invite you to submit a revised version of the manuscript that addresses the points raised during the review process.

The main issue is that CSP duration by SEMG was only 46.4 ms (mean), whereas in many previous studies the CSP duration were over 100 ms and in some cases over 200 ms. Previous studies already demonstrated potential “spinal” inhibition in the first 50 ms of the SP and your study adds further information on this early phase of inhibition. However, it does not provide information to the later stages of CSP (~100 ms). There is some discussion in the paper of the low intensity used but this largely relates to technical issues of not able to test higher intensities with the methods used. It appears unlikely that the mechanisms proposed in the present study can account for the much longer CSP durations reported in other studies in the literature. This need to be clearly acknowledged in the abstract and in the discussion.

The abstract indicates that “the CSP may denote a continuation of the excitatory period initiated by TMS-induced MEP”. While this is technically correct, it is confusing as it suggest that the excitability during CSP is increased but there is decreased number of motor units firing possibly due to the refractory period. Part of the CSP may be related to the falling phase of net EPSP induced by TMS should be mentioned in the abstract.

Please discuss the proposed mechanisms for increased firing rate for the small number of units that fires during the CSP. The statement that “the silent period may not represent a genuine inhibitory period” is an overstatement because there is other evidence for cortical inhibition for example from epidural recording of D and I waves as noted by one of the reviewers.

We would appreciate receiving your revised manuscript by Sep 20 2019 11:59PM. To enhance the reproducibility of your results, we recommend that if applicable you deposit your laboratory protocols in protocols.io, where a protocol can be assigned its own identifier (DOI) such that it can be cited independently in the future. For instructions see: http://journals.plos.org/plosone/s/submission-guidelines#loc-laboratory-protocols

We look forward to receiving your revised manuscript.

Kind regards,

Robert Chen

Academic Editor

PLOS ONE

Journal Requirements:

1. We note that you have indicated that data from this study are available upon request. PLOS only allows data to be available upon request if there are legal or ethical restrictions on sharing data publicly. For information on unacceptable data access restrictions, please see http://journals.plos.org/plosone/s/data-availability#loc-unacceptable-data-access-restrictions.

Reviewers' comments:

Reviewer's Responses to Questions

**Comments to the Author**

1. Is the manuscript technically sound, and do the data support the conclusions?

Reviewer #1: Partly

Reviewer #2: Partly

2. Has the statistical analysis been performed appropriately and rigorously? 

Reviewer #1: Yes

Reviewer #2: I Don't Know

3. Have the authors made all data underlying the findings in their manuscript fully available?

Reviewer #1: Yes

Reviewer #2: No

4. Is the manuscript presented in an intelligible fashion and written in standard English?

Reviewer #1: Yes

Reviewer #2: Yes

5. Review Comments to the Author

Reviewer #1: This study used three different techniques to analyze the same set of data with recordings after a transcranial magnetic brain stimulation. The three techniques were surface electromyography, peristimulus time histogram and peristimulus frequencygram. The recording was a motor evoked potential followed by a cortical silent period in the tibialis anterior or abductor pollicis brevis muscle. The main result was that the well-known suppression of background discharge during silent period cannot be observed with the frequencygram. Instead, the firing rate analysis revealed a long-lasting excitatory period and a long-latency inhibitory period starting at the middle of silent period seen in the surface electromyographic recording. The conclusion was that the silent period may

denote a continuation of the excitatory period initiated by the motor evoked potential. Rebound activity may represent tendon organ inhibition induced by muscle contraction due to the magnetic stimulation or long-latency intracortical inhibition. This is an interesting study with a conclusion different from many (almost all) other studies in the field. I have a few minor comments, mainly for the interpretation of the results.

The authors’ previous studies investigated how the electromyographic recordings in a muscle might estimate the synaptic potential and how surface recording and peristimulus time histogram may lead to the error in the estimation. I think these background knowledges should be briefly reviewed in the introduction. The point should also be discussed with the present results.

Similarly, it was missed in the method part how the technique with peristimulus frequencygram is performed and how it is technically different from the peristimulus time histogram.

One technical issue is that this study used very low stimulus intensity. Therefore, only motoneurons with low firing threshold were recorded. The widely accepted GABAB mediated inhibition during silent period may act on the majority of the corticospinal neurons but not on this group of neurons with low threshold. I wonder if this should be further considered.

The mechanism with the function of tendon organ inhibitory interneuron was extensively discussed. However, I feel the discussion was speculative as the muscle contraction with the very low stimulus intensity should be subtle. By the way, the model illustrated in the last figure was somewhat different from the discussion and could be removed entirely.

Minor points:

Current orientation of the stimulation for the tibialis anterior muscle should be mentioned.

I doubt the 4 cm distance between two electrodes for recording in the abductor pollicis brevis muscle.

Stimulus intensity related to the maximal device output should be reported for both muscles.

Force output related to the maximal output should also be reported.

Reviewer #2: The authors hypothesise that “the CSP may not reflect an inhibitory postsynaptic potential (IPSP) only but include a period of excitatory postsynaptic potential (EPSP)”. In the discussion they suggest that “that the silent period may not represent a genuine inhibitory period, instead it may be due to a falling phase of a compound net EPSP generated by the TMS”. The rebound phase would be related to “twitch-induced autogenic inhibition by tendon organ inhibitory interneurons”.

While the CSP is likely composed of cortical and spinal phenomena, I think the interpretation here doesn’t sufficiently acknowledge the wealth of previous evidence of a cortical IPSP. If one is to reconceptualise the CSP, how do the authors account for the finding that epidural volleys indicate that cortical output is reduced from 50-200ms (Chen et al., 1999)? This is critical, because it is a direct demonstration of reduced cortical output during this time. Peripheral observations on the other hand can only comment on the net effects of cortical, spinal (GTO inhibition etc.) and local phenomena in muscle units. Similarly, TMS-EEG studies have demonstrated inhibitory correlates of the CSP. This evidence needs to be included and interpretation adjusted accordingly. Certainly, as the authors note, pharmacological studies elicit systematic changes in GABA - but what of the study by Pierantozzi et al., 2004 which concluded that the observed effects were not driven by peripheral effects of the drug? Such findings suggest that the CSP is likely related to long lasting IPSP at the cortex. These findings represent just some of the wealth of evidence for cortical inhibitory contributions to the CSP. The authors could perhaps refocus their paper to better acknowledge this evidence and then the additional information that is provided by their measures. The authors work has value in providing further evidence that the CSP may be a messy measure that is confounded by spinal contributions.

6. PLOS authors have the option to publish the peer review history of their article (what does this mean?). If published, this will include your full peer review and any attached files.

Reviewer #1: No

Reviewer #2: No

---

## [Author Response · Author response to Decision Letter 0]

3 Sep 2019

Response to Reviewers

TRANSCRANIAL MAGNETIC STIMULATION INDUCED SILENT PERIOD AND REBOUND ACTIVITY RE-EXAMINED

Reviewer #1

1. The authors’ previous studies investigated how the electromyographic recordings in a muscle might estimate the synaptic potential and how surface recording and peristimulus time histogram may lead to the error in the estimation. I think these background knowledges should be briefly reviewed in the introduction. The point should also be discussed with the present results.

Please see the second and third paragraphs of the Introduction. Also, please refer to the second paragraph of Discussion in the “Cortical silent period may represent a net excitatory postsynaptic potential” section where we discussed the current findings in the light of the previous studies.

2. Similarly, it was missed in the method part how the technique with peristimulus frequencygram is performed and how it is technically different from the peristimulus time histogram.

Please see the Methods “Data processing” section, first and second paragraphs are extended with the information requested.

3. One technical issue is that this study used very low stimulus intensity. Therefore, only motoneurons with low firing threshold were recorded. The widely accepted GABAB mediated inhibition during silent period may act on the majority of the corticospinal neurons but not on this group of neurons with low threshold. I wonder if this should be further considered.

At high stimulus intensities, the stimulus generated large field potentials at latencies where MEP and CSP responses were normally observed. These field potentials are made up of synchronously activated several motor unit potentials hence making it impossible to distinguish selected unit potentials from the field potentials. Therefore, using relatively smaller TMS intensities is a technical limitation in this study. This statement has been added to the Limitations section in the Discussion along with the statement about corticospinal neurons.

4. The mechanism with the function of tendon organ inhibitory interneuron was extensively discussed. However, I feel the discussion was speculative as the muscle contraction with the very low stimulus intensity should be subtle. By the way, the model illustrated in the last figure was somewhat different from the discussion and could be removed entirely.

The discussion about the tendon organ is shortened and the last figure is updated in line with the discussion.

5. Minor points:

Current orientation of the stimulation for the tibialis anterior muscle should be mentioned.

A brief statement has been included. Please see the Materials and Methods Section, Page 8, the first paragraph.

I doubt the 4 cm distance between two electrodes for recording in the abductor pollicis brevis muscle.

It was about 2 cm for APB muscle and 4 cm for TA, section is updated (Page 6, the second paragraph). 

Stimulus intensity related to the maximal device output should be reported for both muscles.

Please see the Materials and Methods Section, Page 10, subheading: stimulation.

Force output related to the maximal output should also be reported.

Updated. Please see the Materials and Methods Section, Page 11, the first paragraph.

Reviewer #2

1. While the CSP is likely composed of cortical and spinal phenomena, I think the interpretation here doesn’t sufficiently acknowledge the wealth of previous evidence of a cortical IPSP. If one is to reconceptualise the CSP, how do the authors account for the finding that epidural volleys indicate that cortical output is reduced from 50-200ms (Chen et al., 1999)? This is critical, because it is a direct demonstration of reduced cortical output during this time. Peripheral observations on the other hand can only comment on the net effects of cortical, spinal (GTO inhibition etc.) and local phenomena in muscle units. Similarly, TMS-EEG studies have demonstrated inhibitory correlates of the CSP. This evidence needs to be included and interpretation adjusted accordingly. 

The discussion is now extended with the information about the previous cortical IPSPs together with epidural recording reference of Chen et al. 1999 and some others. The net EPSP during the CSP and late-onset IPSP (referred to as rebound activity) is now updated with the information stating that both the CSP and rebound activities are mixture of excitations and inhibitions originating from both spinal and supraspinal circuits. Several paragraphs in the Discussion.

2. Certainly, as the authors note, pharmacological studies elicit systematic changes in GABA - but what of the study by Pierantozzi et al., 2004 which concluded that the observed effects were not driven by peripheral effects of the drug? Such findings suggest that the CSP is likely related to long lasting IPSP at the cortex. These findings represent just some of the wealth of evidence for cortical inhibitory contributions to the CSP. The authors could perhaps refocus their paper to better acknowledge this evidence and then the additional information that is provided by their measures. The authors work has value in providing further evidence that the CSP may be a messy measure that is confounded by spinal contributions.

We updated the entire Discussion between Pages 22 and 29 in line with the pharmacological studies, indicating the complex structure of CSP which is a net EPSP followed by a net IPSP, the latter possibly due to the mechanisms of intracortical GABAB mediated inhibition contributed by tendon-organ induced autogenic inhibition and other mechanisms.

Academic Editor

1. The main issue is that CSP duration by SEMG was only 46.4 ms (mean), whereas in many previous studies the CSP duration were over 100 ms and in some cases over 200 ms. Previous studies already demonstrated potential “spinal” inhibition in the first 50 ms of the SP and your study adds further information on this early phase of inhibition. However, it does not provide information to the later stages of CSP (~100 ms). There is some discussion in the paper of the low intensity used but this largely relates to technical issues of not able to test higher intensities with the methods used. It appears unlikely that the mechanisms proposed in the present study can account for the much longer CSP durations reported in other studies in the literature. This need to be clearly acknowledged in the abstract and in the discussion.

We propose that the CSP is a net EPSP made up of the falling phase of the MEP-induced EPSP and IPSPs originating from cortical / spinal circuitries. Our study however does not provide information about these effects individually. Similarly, the so-called rebound activity seems to be a net IPSP possibly made up of GABAB mediated cortical inhibition (as shown by previous studies such as by epidural recording) and/or autogenic inhibition. However, due to field potential generation by high intensity stimulation, we technically could not study the long CSPs as most of the literature talks about. We have now stated these restrictions in the Limitations section of discussion as well as in the abstract.

2. The abstract indicates that “the CSP may denote a continuation of the excitatory period initiated by TMS-induced MEP”. While this is technically correct, it is confusing as it suggest that the excitability during CSP is increased but there is decreased number of motor units firing possibly due to the refractory period. Part of the CSP may be related to the falling phase of net EPSP induced by TMS should be mentioned in the abstract.

An action potential can occur during any part of the TMS induced EPSP, this can be during the rising phase or the falling phase. Once it occurs, however, we do not expect another action potential until the end of that cell’s after-hyperpolarization duration (normal discharge after one inter-spike interval). Therefore, we can suggest that the reduced number of action potentials are not due to refractory period but is due to the phase advancement of spikes. Spikes that were to occur during the ‘silent period’ phase advanced as a result of the extra excitation on the cell’s depolarization trajectory, hence generating a gap after the rising phase of the EPSP (as represented in Figure 6). 

We have updated the statement mentioned by the editor in the Abstract: “We propose that part of the CSP may relate to the falling phase of net excitatory postsynaptic potential induced by TMS”

3. Please discuss the proposed mechanisms for increased firing rate for the small number of units that fires during the CSP. The statement that “the silent period may not represent a genuine inhibitory period” is an overstatement because there is other evidence for cortical inhibition for example from epidural recording of D and I waves as noted by one of the reviewers.

Low number of high discharge rate spikes can only if there is extra excitation on its trajectory. From the well-known current-frequency relationship of neuronal discharge, the time period where spike discharge is higher than normal must represent a net excitation affecting the motoneuron. The reason why the number is low has been illustrated in Figure 6 and its legend: “Rising phase of the EPSP crosses firing threshold at most cases as it is larger than the synaptic noise and also is rapidly-rising. Threshold crossing by rapidly-rising phase of EPSP effectively brings action potentials that were to occur later to an earlier time (phase advance of spikes). This creates a period of low firing probability (cortical silent perriod; CSP) immediately after the rising phase of EPSP as spikes that were to fire in that period moved to occur earlier, generating MEP in the SEMG. Threshold crossing during falling phase of an EPSP is only possible when fast-rising phase fails to cross the threshold and the falling phase of the EPSP crosses the threshold with the help of an up-going synaptic noise. This is an extremely rare event and hence most of the threshold crossings will be achieved during the rising phase of an EPSP especially when EPSP is large.”

This study illustrated the complexity of the mechanisms underlying MEP, CSP and rebound activity which are confounded by the activation of several networks both in the cortical and spinal levels. What we record from the motor units as the final end product is a combined net effect of all the TMS-evoked spinal and supraspinal circuitries. This statement is now included just before the Limitations.

---

## [Decision Letter · Decision Letter 1]

29 Oct 2019

PONE-D-19-18148R1

TRANSCRANIAL MAGNETIC STIMULATION INDUCED SILENT PERIOD AND REBOUND ACTIVITY RE-EXAMINED

PLOS ONE

Dear Prof. Turker,

Thank you for submitting your manuscript to PLOS ONE. After careful consideration, we feel that it has merit but does not fully meet PLOS ONE’s publication criteria as it currently stands. Therefore, we invite you to submit a revised version of the manuscript that addresses the points raised during the review process.

Please review comments and suggestions from Reviewer 2. The statement that the study “does not provide information about later parts of much longer CSPs induced by high intensity TMS” is now included in the Abstract. I suggest that you include a similar statement in the limitation section of the Discussion

We would appreciate receiving your revised manuscript by Dec 13 2019 11:59PM. To enhance the reproducibility of your results, we recommend that if applicable you deposit your laboratory protocols in protocols.io, where a protocol can be assigned its own identifier (DOI) such that it can be cited independently in the future. For instructions see: http://journals.plos.org/plosone/s/submission-guidelines#loc-laboratory-protocols

We look forward to receiving your revised manuscript.

Kind regards,

Robert Chen

Academic Editor

PLOS ONE

Reviewers' comments:

Reviewer's Responses to Questions

**Comments to the Author**

1. If the authors have adequately addressed your comments raised in a previous round of review and you feel that this manuscript is now acceptable for publication, you may indicate that here to bypass the “Comments to the Author” section, enter your conflict of interest statement in the “Confidential to Editor” section, and submit your "Accept" recommendation.

Reviewer #1: All comments have been addressed

Reviewer #2: (No Response)

2. Is the manuscript technically sound, and do the data support the conclusions?

Reviewer #1: (No Response)

Reviewer #2: Yes

3. Has the statistical analysis been performed appropriately and rigorously? 

Reviewer #1: (No Response)

Reviewer #2: Yes

4. Have the authors made all data underlying the findings in their manuscript fully available?

Reviewer #1: (No Response)

Reviewer #2: Yes

5. Is the manuscript presented in an intelligible fashion and written in standard English?

Reviewer #1: (No Response)

Reviewer #2: Yes

6. Review Comments to the Author

Reviewer #1: (No Response)

Reviewer #2: The authors have incorporated many of the reviewer suggestions. I felt that the importance of some of the points raised by reviewers wasn’t fully reflected in the updated manuscript. In particular, the following inclusion just glosses over this point “Previously proposed mechanisms for the cortical silent period have focused on theories regarding cortical inhibition [7, 10, 45, 46].” Actually, I think that in the spirit of transparent/clear science, it is absolutely critical to more clearly and explicitly acknowledge that cortical inhibitory mechanisms contribute to the late CSP, in particular, it should be specifically stated that there is clear evidence that epidural volleys are reduced in amplitude during the CSP (Chen et al, 1999) and that Pierantozzi et al., 2004 concluded that the drug effects they observed were not driven by peripheral effects – this indicates that the late cortical component of the CSP is likely an inhibitory phenomenon mediated by GABA. CSP duration is typically in the order of 100-200ms (Chin et al., Brain Res, 2012) and the time-course of spinal contributions has been partially illustrated using brainstem stimulation (Inglhilleri; 1993; JPhysiol). This leaves plenty of scope for the authors to still demonstrate the mechanisms of the spinal contribution and does not detract from this.

I have a few further other suggestions (in astereisks), which in my view clarify an important distinction relative to the bulk of the literature on this topic:

-Title: “Transcranial magnetic stimulation induced *early* silent period and rebound activity re-examined”

-Abstract – “Our aim is to better characterize the *early* CSP phenomena by combining various analysis tools on firing motor units.”

“Discharge rate analysis, however, revealed not three, but just two events with distinct time courses; a long-lasting excitatory period (71.2 ± 9.0 ms for TA and 42.1 ± 11.2 ms for APB) and a long-latency inhibitory period.” - *insert total duration at end of sentence (i.e. around 46ms). The early phase of the CSP is conventionally considered as the first 50ms, whereas the remainder up to several hundred ms is considered mostly cortical. I think the article is still not making this distinction sufficiently clear.

-In the discussion, the following header could be retitled as follows for clarity: “*Early* cortical silent period may represent a net excitatory postsynaptic potential”

“This limitation also restricted us from directly comparing the MEP and CSP sizes and durations with the literature as they have used much stronger stimulus intensities than this study. *This means our analysis was constrained to the first 50ms of the CSP, whereas CSP duration is typically in the order of 100-200ms.*”

7. PLOS authors have the option to publish the peer review history of their article (what does this mean?). If published, this will include your full peer review and any attached files.

Reviewer #1: Yes: Zhen Ni

Reviewer #2: No

---

## [Author Response · Author response to Decision Letter 1]

3 Nov 2019

Response to Reviewers

TRANSCRANIAL MAGNETIC STIMULATION INDUCED EARLY SILENT PERIOD AND REBOUND ACTIVITY RE-EXAMINED

Reviewer #2

The following inclusion just glosses over this point “Previously proposed mechanisms for the cortical silent period have focused on theories regarding cortical inhibition [7, 10, 45, 46].” Actually, I think that in the spirit of transparent/clear science, it is absolutely critical to more clearly and explicitly acknowledge that cortical inhibitory mechanisms contribute to the late CSP, in particular, it should be specifically stated that there is clear evidence that epidural volleys are reduced in amplitude during the CSP (Chen et al, 1999) and that Pierantozzi et al., 2004 concluded that the drug effects they observed were not driven by peripheral effects – this indicates that the late cortical component of the CSP is likely an inhibitory phenomenon mediated by GABA. CSP duration is typically in the order of 100-200ms (Chin et al., Brain Res, 2012) and the time-course of spinal contributions has been partially illustrated using brainstem stimulation (Inglhilleri; 1993; JPhysiol). This leaves plenty of scope for the authors to still demonstrate the mechanisms of the spinal contribution and does not detract from this.

A paragraph to make the contribution of the cortical inhibitory mechanisms to the late CSP clearer was included in line with the suggestions of the Reviewer in Page 26.

-Title: “Transcranial magnetic stimulation induced *early* silent period and rebound activity re-examined”. 

-Abstract – “Our aim is to better characterize the *early* CSP phenomena by combining various analysis tools on firing motor units.”

“Discharge rate analysis, however, revealed not three, but just two events with distinct time courses; a long-lasting excitatory period (71.2 ± 9.0 ms for TA and 42.1 ± 11.2 ms for APB) and a long-latency inhibitory period.” - *insert total duration at end of sentence (i.e. around 46ms). The early phase of the CSP is conventionally considered as the first 50ms, whereas the remainder up to several hundred ms is considered mostly cortical. I think the article is still not making this distinction sufficiently clear.

-In the discussion, the following header could be retitled as follows for clarity: “*Early* cortical silent period may represent a net excitatory postsynaptic potential”

“This limitation also restricted us from directly comparing the MEP and CSP sizes and durations with the literature as they have used much stronger stimulus intensities than this study. *This means our analysis was constrained to the first 50ms of the CSP, whereas CSP duration is typically in the order of 100-200ms.*”

All included.

---

## [Editor Report · Decision Letter 2]

7 Nov 2019

Response to Reviewers TRANSCRANIAL MAGNETIC STIMULATION INDUCED EARLY SILENT PERIOD AND REBOUND ACTIVITY RE-EXAMINED

PONE-D-19-18148R2

Dear Dr. Turker,

We are pleased to inform you that your manuscript has been judged scientifically suitable for publication and will be formally accepted for publication once it complies with all outstanding technical requirements.

With kind regards,

Robert Chen

Academic Editor

PLOS ONE
---

## [Editor Report · Acceptance letter]

19 Nov 2019

PONE-D-19-18148R2 

Transcranial magnetic stimulation induced early silent period and rebound activity re-examined 

Dear Dr. Turker:

I am pleased to inform you that your manuscript has been deemed suitable for publication in PLOS ONE. Congratulations! Your manuscript is now with our production department. 

With kind regards,

on behalf of

Dr. Robert Chen 

Academic Editor

PLOS ONE